# Protocol: Developing a framework to improve glycaemic control among patients with type 2 diabetes mellitus in Kinshasa, Democratic Republic of the Congo

**Jean-Pierre Fina Lubaki**[1,2]*, **Olufemi Babatunde Omole**[1], **Joel Msafiri Francis**[1]

**1** Department of Family Medicine and Primary Care, University of the Witwatersrand, Johannesburg, South Africa, **2** Department of Family Medicine and Primary Healthcare, Protestant University of Congo, Kinshasa, Democratic Republic of the Congo

\* jeanpierrefina@yahoo.fr

**Data Availability Statement:** This research protocol reports no data. It describes the plan for a

## Abstract

In Kinshasa, Democratic Republic of the Congo (DRC), between 68–86% of patients with type 2 diabetes present with poor glycaemic control leading to increased risk of complications and high cost of care. Identifying the factors driving glycaemic control is essential for better management. There is lack of data on factors associated with poor glycaemic control and targeted interventions in the DRC. This study aims to determine the factors associated with type 2 diabetes control and develop an appropriate intervention package in Kinshasa. The study will comprise of three sub-studies as follows: the first sub-study being a concurrent parallel mixed-methods cross-sectional study to determine factors driving poor glycaemic control among patients in Kinshasa. A total of 614 patients will be invited to participate in a cross-sectional study and respond to standardized questionnaires. A minimum of 20 purposively selected patients will participate in the qualitative study that will involve in-depth interviews about their perspectives on glycaemic control. In the quantitative study, multivariable logistic regression will be performed to determine factors associated with glycaemic control, after identifying the confounding factors. In the qualitative study, thematic analysis will be performed. Findings of the quantitative and qualitative studies on factors that are associated with glycaemic control will be triangulated. And allow to conduct the second sub-study, a qualitative inquiry with a minimum of 20 healthcare providers and 20 patients, selected purposively, to explore their perspectives about potential interventions to improve glycaemic control. At the last, the findings of both sub-studies will be subjected to an anonymous electronic three-round process Delphi study involving 25 stakeholders on the intervention package to develop a framework to optimise glycaemic control in Kinshasa. The implementation of the intervention package will occur after the completion of this study with expected substantial impact on the patients, healthcare providers, and health system.

PhD research protocol with three sub-studies to be completed successively over four years. For these studies, the modalities to access data underlying the results will be indicated at the time of publication. We will favour public access at the Publisher's site.

**Funding:** The authors received no specific funding for this work.

**Competing interests:** The authors have declared that no competing interests exist.

# Introduction

Non-communicable diseases (NCD) are responsible for more than 70% of deaths worldwide, and more than three-quarters of these occur in low- and middle-income countries [1]. Diabetes is one of the major non-communicable diseases. Globally, from 1980 to 2014, the prevalence of diabetes has almost quadrupled [2], and in 2019, it was the cause of 4.2 million deaths and the reason for approximately $760 billion in global health expenditure [3]. Diabetes requires many adjustments in the lives of patients to achieve control which can reduce or even delay the onset of complications [4]. Patients are required to eat healthy, exercise, stop smoking, manage stress, drastically limit alcohol consumption, follow the scheduled visits, adhere to treatment, and participate actively in the treatment process [5]. Worldwide, diabetes control remains a major challenge for people living with the disease with, the proportion of patients with type 2 diabetes achieving glycaemic control being around 50% [6]. In Sub-Saharan Africa, poor glycaemic control rates are higher [7]. Knowledge of the determinants of poor glycaemic control is essential for planning interventions to ensure adequate management [7, 8]. Nevertheless, the reasons for poor glycaemic control in type 2 diabetic patients are complex and can involve factors related to the patient (sociodemographic, economic, biologic, and psychologic factors), caregiver (therapeutic inertia, poor training), the healthcare system (preparation for chronic care model, few resources, organization), and society (culture and environment) [9, 10]. In addition, it should be noted that the presence and importance of these factors vary through locations [7, 10].

This study will be conducted in the Democratic Republic of the Congo (DRC). In 2020, the International Diabetes Federation estimated the prevalence of diabetes in the DRC to be approximately 4.8% [11], with about 92% being type 2 diabetes [12]. The guidelines from the Ministry of Health in the Democratic Republic of the Congo defined glycaemic control as a level of glycosylated haemoglobin < 7.0% [12]. The rate of glycaemic control among diabetic patients is extremely low with–local studies finding that control was not achieved in 68% to 86% of patients [13, 14]. This is within a context of socioeconomic deprivation, where patients must pay from their own pockets and without a good integration of diabetes care in the health system [15]. The burden of diabetes is a real hindrance to the wellness of both patients and their families, and the effectiveness of the health system oriented towards other priorities with limited resources [13], and the burden is greater when glycaemic control is not achieved, and complications arise. It is imperative that all the stakeholders in diabetes management agree to move towards the goal of better glycaemic control [8, 13]. Data on glycaemic control in the DRC are very scarce [12] and the factors associated with glycaemic control are not fully known, and no framework exists to improve control among patients with type 2 diabetes mellitus. Few studies have attempted to analyse factors related to poor glycaemic control among patients with type 2 diabetes and have been conducted in single health facilities. In Kinshasa, Longo-Mbenza et al. found, in a cross-sectional survey in one clinic in Kinshasa, that factors associated with poor control in diabetes were Type 1 diabetes, diabetes duration ≥ 4 years, female gender, underweight, diabetic retinopathy, diabetic nephropathy, elevated total cholesterol and higher levels of HDL-cholesterol [13]. In the latest study to date, Blum et al., in a cross-sectional survey in a centre for the follow-up of diabetic patients in a rural area, were unable to identify any socio-economic factor underlying the very low proportion of glycaemic control [14]. Very little attention was paid to identifying the lived experiences and perspectives of diabetic patients, leading to a gap in knowledge that could have helped the understanding and response to the poor glycaemic control in our setting [16]. Our approach to determine the factors that explain poor glycaemic control will consist of a concurrent parallel mixed methods cross sectional analytic study, which will include assessment of psychological and behavioural

aspects, often neglected in the previous studies, but also of food insecurity, which is an increasingly important factor in the control of diabetes [17]. The identification of the interventions to improve glycaemic control will comprise of two components. The first component is to identify patients and healthcare provider's perspectives on ways to improve glycaemic control. The patients and healthcare providers' perspectives have been found essential to build an adequate system of care for diabetes [18]. These perspectives are variable according to settings and may be different from patients to healthcare providers [19, 20]. In the DRC, to the best of our knowledge, there have been no studies devoted to identifying patients and/or perspectives on the control of type 2 diabetes. Consequently, we have designed a descriptive study on the perspectives of patients and healthcare providers on ways to improve glycaemic control in the DRC. The second component of identifying the interventions will be to search for a consensus from all the stakeholders involved in the management of diabetes in the country and in international level. In Kinshasa, a recent study in Kinshasa Primary HealthCare Network found that the practice of type 2 diabetes care was poor [21]. In the DRC, very few interventions are specifically directed towards glycaemic control [12]. Models for chronic care applied to diabetes recognise the importance of restructuring health systems with greater empowerment of patients, involvement of their families and the communities in the care, need for resources and necessary changes in policies [22, 23]. As highlighted in the conceptual model resumed in the Fig 1, innovative interventions that consider the specificities and context of care for patients with diabetes are essential. The development of this consensus will advance with the identification of factors associated with control of type 2 diabetes in the DRC, and the perspectives of patients and healthcare providers on ways to improve glycaemic control.

## Materials and methods

The study has obtained ethical clearance from the Ethics Committee of the Protestant University of the Congo (Comité d'éthique de l'Université Protestante au Congo, reference number: CEUPC 0067; Date: 05/02/2021) and the Human Research Ethics Committee (Medical) of the University of the Witwatersrand (reference number: M210308; Date: 26/08/2021). This study protocol is registered on Open Science Framework (OSF) since May 1, 2021, with number: osf.io/6fvt4. It will comprise, in addition to a systematic review and meta-analysis on glycaemia control in sub-Saharan Africa (PROSPERO CRD 42021237941), the following three sub-studies: a concurrent parallel mixed methods cross-sectional analytic study on factors for glycaemic control among patients with type 2 diabetes in Kinshasa, a qualitative descriptive study of perspectives from patients and healthcare providers on ways to improve glycaemic control, and a Delphi study to develop an intervention package for glycaemic control. The two first sub-studies will gather the elements for the conceptualisation and choice of interventions in the ultimate phase, the Delphi study. The research plan is summarized in the Fig 2. The cross-sectional study will be reported according to the Strengthening the reporting of observational studies in epidemiology (STROBE) guidelines [24], and the qualitative components of this study will be reported according to Consolidated criteria for reporting qualitative research (COREQ) guidelines [25].

### Sub study 1

This study will be of a concurrent parallel mixed methods cross sectional analytic design to identify drivers of poor glycaemic control among patients with type 2 diabetes mellitus. The key outcome in this study is glycaemic control, which is defined as controlled if the HbA1c is < 7% and uncontrolled if the HbA1c is $\geq$ 7% [12].

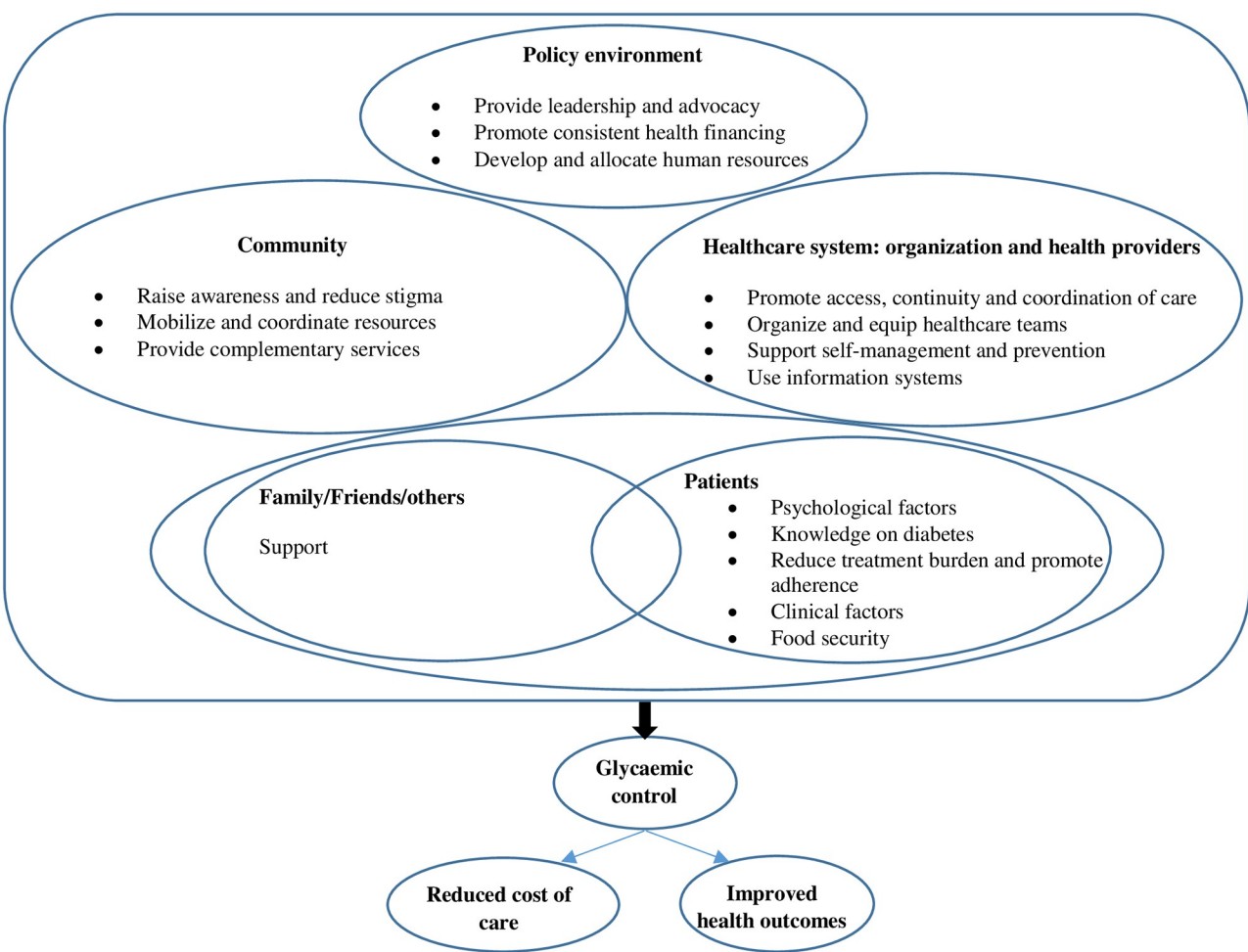

**Fig 1. Conceptual model for glycaemic control among patients with type 2 diabetes mellitus in Kinshasa, DRC.**

**Study setting.**    This study will be multicentric within the city of Kinshasa, the capital of the DRC, which has an estimated total population of about 15 million inhabitants spread over an area of 9,965 km$^2$ [26]. The study will be conducted in the health facilities belonging to the Catholic Church and the Salvation Army. With a total of 66 health facilities (1 referral hospital and 65 health centres) distributed across 24 health districts, these organizations own most of the facilities that have integrated diabetes care in primary care. These healthcare facilities have registered 9170 patients in 2020. The health centres serve as the first point of contact for the patients; these are mainly managed by nurses who ensure routine care and refer patients with problems to the doctor. In the referral hospital in general and in a minority of health centres, outpatient diabetes care is managed by doctors assisted by nurses.

**Qualitative phase.**    This will be a qualitative exploratory study on the lived experiences and perspectives of type 2 diabetic patients on factors driving poor glycaemic control.

*Study population*, *sample size and sampling.* The study population will comprise of patients with type 2 diabetes mellitus followed in the 20 healthcare facilities selected for the study in Kinshasa. These healthcare facilities will be the same selected for the quantitative phase of this sub-study; these will be selected from the total of 66 healthcare through sampling proportional to number of diabetic patients followed. The inclusion criteria will be age ≥18 years and being

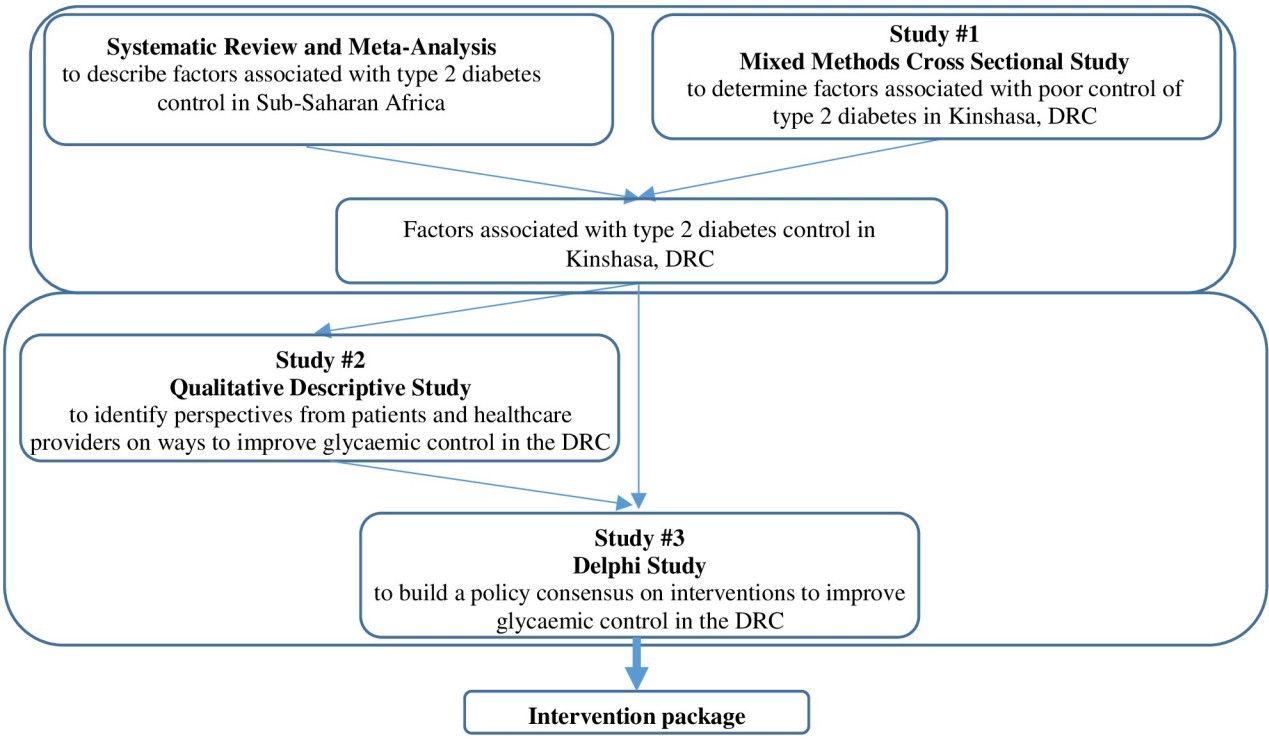

**Fig 2. Research plan.**

on drug therapy for at least six months; the exclusion criteria will be: pregnancy, difficulty communicating due to mental disability, and refusal to give consent.

Previous studies have recommended 20 interviews to be adequate to reach thematic saturation when using thematic analysis, defined as the point at which no new concepts emerge from subsequent interviews [27]. Nevertheless, heterogeneity of participants and the complexity of the glycaemic control in diabetes will be considered in purposively recruiting patients e.g., ensuring diversities in sex, duration of illness, socioeconomic strata, treatment categories, good and poor glycaemic control, are represented. While expecting to recruit a minimum of 20 patients, data saturation will determine the ultimate sample size.

We plan to purposively select patients with type 2 diabetes with both good and poor glycaemic control, based on their propensity to report their experiences. Health facilities' teams who know the patients will help in selecting them; these patients will then be invited for interviews in the healthcare where they receive diabetes care.

*Data collection.* Prior to the commencement of sub study 1, the Principal Investigator will recruit the research assistants (four medical doctors) and train them on study procedures, research ethics and will ensure all have undertaken the online research ethics course. Research assistants will be responsible for administrative tasks and recruiting participants. Prior to the interviews, the Principal Investigator will provide study information summarized on a leaflet to potential study participants and obtain written informed consent. A copy of the leaflet and that of the signed consent form (to participate in the study and for audio recording) will be given to the participant. In the respective facilities selected for the study, the interviews will be conducted by the Principal Investigator in a quiet room where the confidentiality of the participants' statements will be guaranteed. The interviews will be guided by an interview schedule (S1 Appendix). The interviews will be conducted in French or in Lingala depending on the

preferences of the participants. All interviews will be audio recorded. A maximum of three patients are expected to be interviewed per day for three times a week. It is envisaged that data collection for the qualitative study will be completed in about 3 weeks. Each in-depth interview will take approximately at most 45 minutes. At the end of the interview, a sample of 2 millilitres of veinous blood will be drawn from patients for glycosylated haemoglobin assay. We will report on the distribution of poor versus good glycaemic controlled patients in the sample. Data collection will be held in strict respect of rules edited for the prevention of the COVID-19 infection in the country and implemented in the selected health facility.

*Data management and analysis*. The recorded interviews will be transcribed verbatim and then translated into English language by a language expert for analysis. Data coding and thematic analysis will be performed using the MAXQDA version 20. Similar themes would be coalesced into major themes, and similar major themes, if necessary, will be aggregated into categories. Outlying lived experiences will be sought out for and explanatory model that best explain the inter-relatedness of themes and categories, and the experiences of patients on their own glycaemic control.

*Credibility, transferability, dependability, and confirmability*. The credibility of the study will be enhanced by peer debriefing and member checking; thick description of data will ensure transferability; external audits conducted by the supervisors and one external expert on qualitative research will ensure dependability, and the confirmability will be ensured by external audits and audit trial [28]. Triangulation of findings between the principal investigator and the supervisors will enhance credibility.

*Bias*. The biases that may interfere with the data are as follows: selection, interviewer, and social desirability biases. The Principal Investigator will ensure that a comfortable environment has been provided for the interviews and ensure confidentiality to participants to minimize the potential biases.

**Quantitative phase.** This will be a cross-sectional analytic study among patients with type 2 diabetes mellitus attending at the health centres and hospitals for type 2 diabetes mellitus in Kinshasa, DRC.

*Study population and setting*. The study will be conducted in selected healthcare facilities organizing diabetes care in Kinshasa. The total number of patients registered in these healthcare structures is 9170. The inclusion criteria will be age $\geq$18 years, and on drug therapy for at least six months; the exclusion criteria will be: pregnancy, difficulty communicating due to mental disability, and refusal to give consent.

*Sample size estimation for the cross-sectional study*. The estimated minimum sample size has been computed by Epi info version 7.2.2.2 assuming that the prevalence of poor glycaemic control is 68% [13], and a 95% confidence level, a power value of 80%, that among patients who have been on diabetes treatment for $\leq$ 7 years (Unexposed), 59,2% present with poor glycaemic control and those who have been on treatment for **>** 7 years (Exposed), 74,4% present with poor glycaemic control, and a ratio (Unexposed: Exposed) of 0.47 [29]. The minimum estimated sample size is 368. Adjusting for a design effect of 1.5, the calculated sample size of 552 was determined. At the last, to account for 10% non-response rate, the minimum required sample size is 614 patients.

*Sampling and recruitment of participants*. As the healthcare facilities have unequal number of patients, we will select the participants through probability proportional to the patient population size. Through this strategy, we plan to randomly select 20 health facilities and in each selected health facility, a minimum of 31 patients will be randomly selected. This process will ensure that as result, a sample where each patient has the same overall probability of selection or self-weighting.

Practically, for the selection of the participants, at each selected health facility, the research assistant will recruit eligible (age ≥18 years, on treatment ≥ 6 months) type 2 diabetic patients attending the clinic at the respective clinic day. The research assistant will pick the file of the first patient on a clinic day to assess for eligibility and thereafter will pick every 3rd patient's file for the subsequent three patients for the administration of questionnaire in a particular day. In a situation where the first patient is ineligible or the 3rd picked patient in the subsequent stage is ineligible, the research assistant will pick the next patient/s until he/she obtains an eligible patient and then continue with every 3rd patient selection. If a patient refuses to participate, the research assistant will pick the next patient on the list and thereafter, pick every 3rd patient on the list. The process will continue until the number of patients is reached. We intend to recruit about 4 patients per each of the five data collectors a day three times a week for a maximum of 11 weeks until the minimum sample size is attained.

*Data collection*. The research team (principal investigator and research assistants) will provide study information to each potential study participant and then obtain a written informed consent. Data collection will be held in strict respect of rules edited for the prevention of the COVID-19 infection in the country and implemented in the selected health facility.

*The administration of the questionnaire*. Once a patient is declared eligible and gave consent for the study, the research team will proceed to assess physical and anthropometric measurements. Then, the data collectors will administer the questionnaires. The administration of the questionnaire will last approximately up to 60 minutes. At the end of the data collection, a blood sample is taken from the participant. The research team (principal investigator and research assistants) will administer the questionnaire to the study participants using an electronic questionnaire prepared in REDCap on a Tablet/Smartphone. The questionnaire will be administered in French or in Lingala depending on the preferences of the participants. The information that will be sought from the participants will be the followings: patient factors (sociodemographic factors (age, gender, ethnic group, marital status, occupation, household income, education level, number of people with diabetes in the household, social support), psychological factors (self-care management, diabetes distress, depression), knowledge on diabetes, lifestyle (smoking, alcoholism, practice of exercise)); degree of food insecurity (food security, low food security, very low food insecurity); clinical factors (physical measurements (weight, height, body mass index, waist circumference), characteristics of diabetic disease (family history, duration, number and reason for previous hospitalisations, co-morbidity (hypertension, depression, obesity. . .), treatment. . .); health system factors (existence of health insurance, distance from health centre). The information about the characteristics of diabetes disease will be verified in the patients' medical records, if available.

Measurements tools and operational definitions.

- Patients with type 2 diabetes will be those identified as such in the medical record.

- Glycaemic control or diabetes control will be defined according to the followings: adequate control—HbA1C <7.0%, poor control—HbA1C ≥7.0% [15, 30].

- A distance from participant's place of residence to the health centre of **< 5 kilometres** or **< 1 hour** will be considered as adequate while **> 5 kilometres** or more than 1 hour will be inadequate.—A daily income < $1.9 will be set as threshold for poverty [31].

- Alcohol use will be assessed with the Alcohol Use Disorders Identification Test (AUDIT-C) [32]; a score ≥ 4 (in men) or ≥3 (in women) represents health-risk consumption while a score ≥ 5 (in men) or ≥ 4 (in women) indicates alcohol dependence.

- Nicotine dependence among smokers will be assessed with the Fagerstrom Test for Nicotine Dependence (FTND) [33]; the interpretation of the score obtained is as follows: 1–2: low dependence, 3–4: low to moderate dependence, 5–7: moderate dependence, and $\geq$ 8: high dependence.

- Physical activity will be assessed by use of the Global Physical Activity Questionnaire (GPAQ). The GPAQ consists of 16 questions covering three domains as: activity at work, transport and leisure activities [34].

- Trained staff using validated devices will measure clinical parameters (height, weight, waist circumference and blood pressure). Relating to the BMI, patients will be categorized as underweight (BMI<18.5 Kg/m$^2$), normal weighted (18.5–24.99 Kg/m$^2$), overweighed (BMI$\geq$25 Kg/m2), and obese: $\geq$30Kg/m2 [35]. For the waist circumference, normal values are defined according to gender as: men <102 centimetres; women < 88 centimetres [36]. Hypertension will be defined as a systolic blood pressure level of 140 mmHg or higher and/or a diastolic blood pressure level of 90 mmHg or higher [37].

- Food security status will be measured with the household food insecurity scale (HFIAS), a 9-item measurement tool. It reports the experience of past four weeks [38, 39]. The HFIAS categorises households into four levels of food access: food secure, mild food insecure, moderately food insecure and severely food insecure.

- Knowledge of diabetes will be assessed by the use of the simplified diabetes knowledge scale, a true/false response format of the Revised Diabetes Knowledge Scale of the Michigan Diabetes Research Center [40, 41]. It contains 20 items with three related to insulin users. The internal reliability of the Revised Diabetes Knowledge Scale is 0.71 versus 0.61 for the Simplified Diabetes Knowledge Scale; the item correlation with the total knowledge test score ranged from 0.23 to 0.45 for the Revised Diabetes Knowledge Scale versus 0.26–0.58 for the Simplified Diabetes Knowledge Scale. The corrected item-total correlations were >0.2 for the Revised Diabetes Knowledge Scale and the Simplified Diabetes Knowledge Scale [40]. The Simplified Diabetes Knowledge Scale is used as a continuous score; no cut-off point has been defined for knowledge.

- Adherence to treatment will be assessed by the Morisky Green Levine test (MGL test) [42]. The MGL test is a structured four-point measure of treatment adherence with an alpha reliability of 0.61. The MGL test divides patients into three categories of adherence: high, medium, and low.

- Social support will be assessed by the Multidimensional Scale for Perceived Social Support (MSPSS) [43]; the MSPSS is a brief self-report questionnaire that contains 12 items rated in a five-point Likert-type scale. The MSPSS assessed three sub-scales of the social support as: family, friends, and significant others; its internal consistency and test-retest reliability are excellent (Cronbach's alpha: 0.81 to 0.98 in non-clinical samples and 0.92–0.94 in clinical samples) [44]. For the Multidimensional Scale for perceived social support (MSPSS), the mean scores are calculated as: Significant Other Subscale—sum across items 1, 2, 5, & 10, and divided by 4. Family Subscale—sum across items 3, 4, 8, & 11, then divided by 4. Friends Subscale—sum across items 6, 7, 9, & 12, then divided by 4. Total Scale—sum across all 12 items and divided by 12. Any mean scale score ranging from 1 to 2.9 could be considered low support, a score of 3 to 5 as moderate support, and a score from 5.1 to 7 high support.

- Screening of depression will be done using the Patient Health Questionnaire-9 (PHQ-9) [45]. PHQ-9 is the 9-point depression module of the full PHQ and is a screening tool and

measure of depression severity. The Cronbach's alpha was 0.89 in a primary healthcare study [46]. Depression screening with PHQ-9 will determine the following categories: moderate depression (12–14), moderately severe depression (15–19), severe depression (20–27) [45].

- Diabetes distress will be assessed through the Diabetes Distress Scale (DDS) [47], a 17-item scale that captures four critical dimensions of distress: emotional burden (5 items), regimen distress (5 items), interpersonal distress (3 items) and physician distress (4 items). Associations have been found between DDS scores and behavioural management and biological variables when DDS scores are > 2.0 [48]. For the DDS, a mean item score 2.0–2.9 should be considered 'moderate distress,' while a mean item score > 3.0 'high distress'.

- Diabetes self-management will be assessed with the use of the Diabetes Self-Management Questionnaire (DSMQ), a 16-item questionnaire recognised to be a reliable and valid instrument that enables an efficient assessment of self-care behaviours associated with glycaemic control [49]. It comprises for subscales: dietary control (4 items), glucose management (5 items), physical activity (3 items), and physician contact (3 items).

*Laboratory procedures*. All patients will have 2 millilitres of venous blood taken by the researchers for determination of glycosylated haemoglobin. The glycosylated haemoglobin will be assayed using the same validated method at one central laboratory at the Didactic Health Centre of the School of medicine at the Protestant University of Congo at Kinshasa to avoid differences between laboratories [50].

*Variables*. Outcome: the main outcome variable will be the glycaemic control that will be assessed with HbA1c. Glycaemic control will be defined as HbA1c <7% [12, 26].

Exposures of interest: potential factors associated with glycaemic control as described above in the information sought from the participants and related to patients, society, and health system (S2 and S3 Appendices).

*Data analysis*. Statistical analysis will be performed using STATA 17. Quantitative variables normally distributed will be expressed as mean ± standard deviation; and non-normally distributed data as median with interquartile range (IQR). The categorical variables will be expressed as frequency (n) and percentage (%). Subsequently, we will perform the bivariate analysis to determine factors associated with glycaemic control using t -test or Mann-Whitney U for continuous variables and Chi square test/Fisher exact test for categorical variables. We will fit a multivariable logistic regression to assess factors associated with glycaemic control. Age, sex, duration on treatment, food security will be included in the regression model a priori. And other factors will be included in the model if they have a p-value of <0.20 in bivariate analysis. The p value of <0.05 will be considered the statistically significant.

*Reliability, validity, and generalisability*. Ensuring the reliability of the study will be using validated questionnaires for data collection and training of the research team. The findings will be generalisable to the DRC context but could be extended for use in similar settings in SSA.

*Bias*. The potential biases for this study are—selection (non- response of the eligible participants), recall, interpretation, and social desirability. We will minimize the biases by ensuring the research assistants are properly trained and the aim of the study is clearly explained to the participants.

## Sub study 2

Using the findings from the sub study 1, we will carry out a qualitative study in the four clinics in Kinshasa to explore patients' and healthcare providers' perspectives on how to improve the control of glycaemia.

**Study population and sample.** Patients and healthcare providers aged ≥18 years will be invited to participate. Patients and healthcare providers will be chosen deliberately. Patients to be selected will be those who have been followed for at least six months for diabetes and who have mastered the model of care offered to patients with diabetes. Caution will be taken to select those who will be willing to discuss their views on improving blood glucose control. Attention will be paid to selecting male and female patients, from different socio-economic categories and ages, and independently of their status regarding glycaemic control (controlled and uncontrolled). About caregivers, preference will be given to selecting providers with long experience of diabetes. Male and female providers will be selected as well as doctors and nurses. The participants will be excluded if they present with difficulty communicating due to mental disability or, refuse to give consent.

For each category of participants, patients, or healthcare providers, as stated by many authors for adequate sample size, we expect to have 20 interviews but will continue to include participants until thematic saturation [27]. The total of expected participants is 40; twenty (20) patients and twenty (20) healthcare providers. Health managers in charge of the healthcare facilities will help in selecting the health providers to be included in the study. In turn, health facilities' teams who know the patients will help in selecting them; these patients will then be invited to come to the health facilities for interviews. The research team will ensure diversity of participants in terms of sex/ gender, duration of diabetes, treatment types, type of healthcare facility, etc. for patients and sex, years of professional experience, clinical category, type of facility, etc. for healthcare providers. A maximum of three participants are expected to be interviewed per day for three times a week. A maximum of five weeks will suffice to complete data collection.

**Data collection.** The data collection will be carried through in-depth interviews which will be audiotaped. Prior to the interviews, the Principal Investigator will provide study information to potential study participants and obtain written informed consent for both the interview and audio recording. The interviews will be guided by an interview schedule (S4 Appendix). This interview schedule will be updated based on the findings from the systematic review associated with this research and the sub-study 1 on factors driving glycaemic control. At the respective facilities, the principal investigator will interview the participants in a quiet room where the confidentiality of the participants' statements will be guaranteed. The interviews will be conducted in Lingala or French depending on the preferences of the participants. All interviews will be audio recorded. Data collection will be held in strict respect of rules edited for the prevention of the COVID-19 infection in the country and implemented in the selected health facility.

**Data management and analysis.** The recorded interviews will be transcribed verbatim and then translated into English language for analysis. Data coding and thematic analysis will be performed using the MAXQDA version 20. Identified themes that are similar would be coalesced into major themes, and similar major themes, if necessary, will be aggregated into categories. Outliers' perceptions will be sought out for. Attempts will be made to develop explanatory models that best explain the inter-relatedness of themes and categories, and the perceptions of patients and healthcare providers on interventions that can improve glycaemic control in type 2 patients.

**Credibility, transferability, dependability, and confirmability.** The credibility of the study will be enhanced by triangulation, peer debriefing and member checking; thick description of data will ensure transferability; external audits conducted by the supervisors and one external expert on qualitative research will ensure dependability, and the confirmability will be ensured by external audits and audit trial [28].

**Bias.**   The biases that may interfere with the data are—selection (non-response of the eligible participants), interviewer, and social desirability biases.

## Sub study 3

This will be a Delphi study, seeking policy consensus from important stakeholders in diabetes management in developing the package of interventions to improve glycaemic control based on the use of findings from the sub studies 1 and 2. A research team lead by the Principal Investigator and including one supervisor and a statistician will conduct data collection and initial analysis. Data interpretation will be done by the whole team of investigators.

**Study site.**   The Delphi study will be anonymous electronic survey. The participants will respond on potential glycaemic control interventions in Kinshasa.

**Study population and sample size.**   The Delphi study will include all representatives of stakeholders in diabetes management in Kinshasa, DRC. Twenty-five people will be involved–five clinicians from the health centres, five managers in diabetes care, five representatives from universities, five representatives from partners in diabetes management, and five representatives from the Ministry of health. The clinicians are persons qualified in clinical practice of medicine being doctors or nurses and taking care of diabetes; the managers are individuals who are in charge of a healthcare structure (hospital, health centre) or a group of healthcare structures organizing diabetes care; the representatives of universities will be diabetologists or internal medicine experts; partners are all the persons or organizations participating in the management of diabetes by providing financial or technical support; the representatives from the Ministry of health will be individuals working in the National Program of diabetes and the National Directorate against diseases. The participants will be purposively selected with caution to ensure motivation for the subject and expertise on diabetes control. Contacts will be made with the structures organizing or providing a support for diabetes care to have delegates as experts in the issue; known experts in the diabetes management will be contacted directly through emails.

**Inclusion/Exclusion criteria.**   All the participants will have an age $\geq$18 years; the exclusion criteria being difficulty communicating due to mental disability, conflict of interest in study participation, and refusal to give consent. Experts will be local or international experts with preference given to those with current or past work in DRC.

**Data collection.**   Participants will receive by e-mail a document on the purpose and objectives of the study. They will be asked to consent electronically to participate in the study. Once they accept to participate in the study, a second e-mail will inform them on the results of previous sub-studies of this study. This Delphi study will be a three rounds process. At the first round, each stakeholder will be approached to answer an open-ended questionnaire on interventions to improve diabetes control in patients with type 2 diabetes. The participants' responses at the first round will be used to identify items that will be used by the researchers to draw a well-structured questionnaire on interventions to improve glycaemic control. In the second round, experts will be subjected to the questionnaire built after the first round and asked individually to rank the interventions in terms of priority. Their ratings will consist of assessing each intervention according to a 4-point Likert scale ('strongly agree', 'agree', 'disagree', 'strongly disagree'). Additionally, they will be asked to motivate their ranking. At the end of the second round, the researcher will summarize the rankings from the experts and their motivations and feed back to the experts in the third round. In this third round, experts will be asked to re-rank after reconsidering their opinions or judgements in light of other group members' rankings or to specify the reasons for remaining outside the consensus. At the end of the third round, the researchers will sort out a list of remaining interventions and their

rankings along with minority opinions and interventions achieving consensus. The experts' contributions will be anonymous. Researcher team' correspondences with experts will be made in French and English while experts' contributions will be made in English or French depending on their preferences; feedback by the researcher will be in English and French.

**Data analysis.** The data from the first round will more consists of qualitative data; the qualitative data will be summarized using content analysis. In the subsequent rounds, for each round, medians will be used to describe group responses to each statement. And, between rounds, data will be analysed to identify convergence and change of experts' opinions or judgements. Consensus will be defined as **>** 70% of experts agreeing/strongly agreeing or disagreeing/strongly disagreeing with a statement in the Round 3. Stability of consensus will be considered as reached if the between round group responses vary by ≤10%.

## Ethical considerations and declarations

The researchers declare that they will comply with the conditions under which this study has obtained approval from the ethics committees of the Protestant University of Congo and Human Research Ethics Committee (Medical) of the University of the Witwatersrand. Any departure from the protocol will need two Ethics Committees. The authorization has been obtained from the Kinshasa Primary HealthCare Network to conduct the study. A written consent will be sought from participants of each of the sub-studies of this research. Withdrawal from the study will have no impact on the care that patients receive or on the position that health care providers occupy within the health care structure. No interventions will be carried out on study participants. During this period of the pandemic, data will be collected in strict compliance with the rules laid down for the protection of individuals in health facilities, namely: wearing masks, social distancing, frequent hand washing and curfews. In this way, crowds will be avoided throughout the data collection process. The research data will be kept at the Protestant University of Congo for a period of 20 years after the end of the study. The participants will not be paid for their participation in the study. Nevertheless, the participants will be reimbursed for the transport to come to the healthcare structure in case they will be invited. If one of the scales used in this study suspects a medical condition for which a participant needs to be managed, s/he will be referred to the appropriate facilities for management.

The research assistants will be acknowledged for their support in the study. The authorship will be based on the fulfilment of all the four conditions listed in the International Committee of Medical Journal Editors uniform requirements for manuscripts submitted to BioMedical Journals [51].

## Status and timeline of the study

The Table 1 presents the timeline of the study. None form of data collection has been conducted at this stage.

## Discussion

Diabetes mellitus, like other NCDs, is growing in importance globally with substantial impact on mortality and a huge economic burden to societies [52, 53]. The progress of diabetes is dramatic in low-and-middle countries, like the DRC, because of the coexisting traditional health priorities, few resources, and the unpreparedness of the health system [8, 54, 55]. The control of glycaemia has shown to prevent or delay the occurrence of the complications [56]. Nevertheless, the control of glycaemia requires many adjustments for all the actors in the diabetes care for better achievements. In the DRC, recent studies have shown high rates of poor control among patients with type 2 diabetes. The burden of diabetes for the patients and the families is

**Table 1. Timing of the activities of the study.**

| | 2020 | | | | 2021 | | | | 2022 | | | | 2023 | | | |
|---|---|---|---|---|---|---|---|---|---|---|---|---|---|---|---|---|
| | Q1 | Q2 | Q3 | Q4 | Q1 | Q2 | Q3 | Q4 | Q1 | Q2 | Q3 | Q4 | Q1 | Q2 | Q3 | Q4 |
| **Protocol drafting and approval** | ■ | ■ | ■ | ■ | ■ | ■ | ■ | | | | | | | | | |
| **Sub study 1** | | | | | | | | | | | | | | | | |
| Data collection | | | | | | | | ■ | ■ | ■ | | | | | | |
| Data analysis | | | | | | | | | ■ | ■ | | | | | | |
| **Sub study 2** | | | | | | | | | | | | | | | | |
| Data collection | | | | | | | | | | ■ | ■ | ■ | | | | |
| Data analysis | | | | | | | | | | | | ■ | | | | |
| **Sub study 3** | | | | | | | | | | | | | | | | |
| Data collection | | | | | | | | | | | | ■ | ■ | ■ | | |
| Data analysis | | | | | | | | | | | | | ■ | ■ | | |
| **Dissemination of the findings** | | | | | | | | | | ■ | ■ | ■ | ■ | ■ | ■ | ■ |

Q1: first quarter, Q2: second quarter, Q3: third quarter, Q4; fourth quarter.

multiplied when complications arise and represents a real obstacle for the wellness of patients and families, and the effectiveness of the health system [57]. As we stated earlier, it is time to act unless we let diabetes continue with its innumerable consequences. In this way, as factors for poor control of glycaemia vary according to settings, the identification of factors acting primer roles in Kinshasa is a pre-requisite if we want effective interventions [7]. From the few studies that have been conducted in the DRC, data on factors driving glycaemic control are scarce [13, 14, 58]. And there were not many efforts to look on how the patients explain by themselves poor glycaemic control that would contribute to the understanding of the phenomenon. Moreover, elements to build an adequate system of care for diabetes are unknown as we have not yet explored perspectives from patients and the healthcare providers on ways to improve glycaemic control.

This study will describe the factors associated with glycaemic control among patients with type 2 diabetes in Kinshasa, DRC. And further explore the perspectives from patients and healthcare providers on ways to improve the control of diabetes. The findings will inform the development of the targeted intervention package to improve glycaemic control in Kinshasa, DRC.

Findings of the planned studies should be interpreted in consideration of the following limitations: the cross-sectional design of the study will not allow to establish the causal inference for factors of poor glycaemic control. The qualitative nature of some components will not guarantee the generalisability of the findings to other settings not similar to Kinshasa, DRC.

## Supporting information

**S1 Appendix. Interview guide for patients with type 2 diabetes sub-study 1.**
(DOCX)

**S2 Appendix. Questionnaire for patients with type 2 diabetes sub-study 1.**
(DOCX)

**S3 Appendix. Laboratory sheet.**
(DOCX)

**S4 Appendix. Interview guide for patients and healthcare providers sub-study 2.**
(DOCX)

## Acknowledgments

The authors acknowledge the team of the Department of Family medicine and Primary Care, and that of that of the School of Public Health of the University of the Witwatersrand for their contributions in the drafting of this study protocol.

## Author Contributions

**Conceptualization:** Jean-Pierre Fina Lubaki.

**Supervision:** Olufemi Babatunde Omole, Joel Msafiri Francis.

**Writing – original draft:** Jean-Pierre Fina Lubaki.

**Writing – review & editing:** Joel Msafiri Francis.

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
