## [Decision Letter · Decision Letter 0]

2 Mar 2022

PONE-D-21-29300

Protocol: Developing a framework to improve glycaemic control among patients with type 2 diabetes mellitus in Kinshasa, Democratic Republic of the Congo

PLOS ONE

Dear Dr. Fina Lubaki,

Thank you for submitting your manuscript to PLOS ONE. After careful consideration, we feel that it has merit but does not fully meet PLOS ONE’s publication criteria as it currently stands. Therefore, we invite you to submit a revised version of the manuscript that addresses the points raised during the review process.

We look forward to receiving your revised manuscript.

Kind regards,

David Desseauve, MD, MPH, PhD

Academic Editor

PLOS ONE

3. Please amend your authorship list in your manuscript file to include author list.

Reviewers' comments:

Reviewer's Responses to Questions

**Comments to the Author**

1. Does the manuscript provide a valid rationale for the proposed study, with clearly identified and justified research questions?

Reviewer #1: Yes

Reviewer #2: Yes

Reviewer #3: Yes

2. Is the protocol technically sound and planned in a manner that will lead to a meaningful outcome and allow testing the stated hypotheses?

Reviewer #1: Yes

Reviewer #2: Yes

Reviewer #3: Yes

3. Is the methodology feasible and described in sufficient detail to allow the work to be replicable?

Reviewer #1: Yes

Reviewer #2: Yes

Reviewer #3: Yes

4. Have the authors described where all data underlying the findings will be made available when the study is complete?

Reviewer #1: Yes

Reviewer #2: Yes

Reviewer #3: Yes

5. Is the manuscript presented in an intelligible fashion and written in standard English?

Reviewer #1: Yes

Reviewer #2: Yes

Reviewer #3: Yes

6. Review Comments to the Author

You may also provide optional suggestions and comments to authors that they might find helpful in planning their study.

Reviewer #1: This paper is about 3 protocols for Developing a framework to improve glycaemic control among patients with

type 2 diabetes mellitus in Kinshasa.

The 3 sub studies are rational, well design and their protocols provied a lot of details.

Depending on the local culture, the authors are may be a little ambitious to count on a rate as low as 10% for non responder for sub study 1.

Reviewer #2: Dear Authors, dear Editor,

Thank you very much for giving me the opportunity to review the present study protocol. The aim of the present three tiers-study is to determine the factors associated with type 2 diabetes management it Kinshasa, RDC.

Although I am not an expert in the field of diabetology , I have several comments regarding the study protocol:

> The abstract does not clearly separate the three aims of this study.

> Introduction: line 32, conducting multivariable logistic regression is ok, but only after having identified confounding factors

> Power calculation ok, Timeline ok but for Table 1, Line 465 please modify the timeline to actual timing

> Introduction line 56-57: be more specific when referring to “setting”

> Study setting line 134: any reference for the number of inhabitant ?

> Line 158: selection of patient for interview can induce an obvious bias. Why not invite randomly or all participants until saturation is reached ?

> Line 230 : Not clear when during the interview procedure all the questionnaires will be administred: Furthermore, there is some concerns that the mulitiplication of questionnaires, the time necessary for completing them and especially the fact that if it is too long the participant might answer inappropriately. How does the author address this specific issue ?

> Line 237, would recommend to add the current working status

> No informations is given regarding who will drive the Delphy consensus data collection and interpretation.

> No information is given regarding the research assistant’s integration in the authorship.

> Is the retribution of the participants specifically specified before inclusion ? It might obviously induce some bias

> How long is an interview planned to last ? Is there a timeframe for the interview ?

Reviewer #3: In Kinshasa, a high percentage of T2DM patients (68%-86%) have poor glycemic control, leading to an increased risk for complications and health care costs. So far, the experiences and perspectives of both diabetic patients and healthcare providers have not been considered in determining: (1) the drivers for poor glycemic control and (2) interventions for improving glycemic control. These factors are explored in this study, together with the input from stakeholders and experts in the field, which makes the study a relevant and innovative topic. Although the manuscript recognizes the importance of restructuring health systems with greater empowerment of patients, involvement of their families and the communities in the care, the perspectives of family members are not taken into account in this study.

There is a high risk for selection biases, which have also been described in the manuscript.

7. PLOS authors have the option to publish the peer review history of their article (what does this mean?). If published, this will include your full peer review and any attached files.

Reviewer #1: No

Reviewer #2: No

Reviewer #3: No

---

## [Author Response · Author response to Decision Letter 0]

29 Mar 2022

RESPONSES TO REVIEWERS’ COMMENTS

Journal requirements

• We note that you have stated that you will provide repository information for your data at acceptance. Should your manuscript be accepted for publication, we will hold it until you provide the relevant accession numbers or DOIs necessary to access your data. If you wish to make changes to your Data Availability statement, please describe these changes in your cover letter and we will update your Data Availability statement to reflect the information you provide. 

 R/ This is a study protocol. It describes the plan of studies that will be conducted and published 

 subsequently until the completion of a PhD degree over four years. The data underlying the 

 findings of the three studies will be published as attachments. A change for the Data availability 

 statement has been mentioned in the cover letter.

 See Cover Letter

• Please amend your authorship list in your manuscript file to include the author list

 R/ The authorship in the manuscript has been amended.

 See Page 1

Reviewers’ comments

Reviewer#1

• Depending on the local culture, the authors may be a little ambitious to count on a rate as low as 10% for non-responders for sub-study 1.

 R/ We thank the reviewer for the valuable comment. We decided on the 10% non-response rate 

 based on the experience from other studies on diabetes in the study setting. For example, studies 

 by Muyer et al in 2012, and Cedrick et al in 2021 had respectively a non-response rate of 6.3% 

 and 4.5% (Muyer et al., 2012, Cedrick et al., 2021). 

 References

 Cedrick, L. M. et al. (2021) ‘Prevalence and determinants of poor glycaemic control amongst 

 patients with diabetes followed at Vanga Evangelical Hospital, Democratic Republic of the 

 Congo, African journal of primary health care & family medicine, 13(1). doi: 

 10.4102/phcfm.v13i1.2664.

 Muyer, M. T. et al. (2012) ‘Diabetes and intermediate hyperglycaemia in Kisantu, DR Congo: 

 A cross-sectional prevalence study’, BMJ Open, 2(6), pp. 1–7. doi: 10.1136/bmjopen-2012- 

 001911.

Reviewer#2

• The abstract does not clearly separate the three aims of this study

 R/ Thanks for this comment. We have now specified the aim for each of the proposed studies.

 See Lines 46-47, 55-57, 57-59

• Line 32, conducting multivariable logistic regression is ok, but only after having identified confounding factors

 R/ Thanks for this comment. In the process described for data analysis, we have planned to 

 conduct bivariate analysis before multivariable analysis and include variables with p<0.20 in the 

 multivariable models. We also specified confounders a priori as age, sex, treatment duration, and 

 food security. We have updated the abstract accordingly to include the planned analyses.

 See Lines 51-52

• Line 465 please modify the timeline to actual timing. 

 R/ We are grateful for this comment. The timeline has been updated.

 See Line 533: Table 1

• Introduction line 56-57: be more specific when referring to “setting”

 R/ Thanks for this comment. We have changed ‘settings’ to ‘locations’ to be more accurate.

 See Line 82

• Study setting line 134: any reference for the number of inhabitants?

 R/ A reference for the number of inhabitants has been included.

 See Line 161 and Reference number 26

• Line 158: selection of patient for interview can induce an obvious bias. Why not invite randomly or all participants until saturation is reached?

 R/ Thanks for this comment. Given the nature of the study design, the intention is not to recruit a representative 

 sample but patients who are information-rich and would be likely to share their perspectives on glycaemic control. 

• Line 230: Not clear when during the interview procedure all the questionnaires will be administered: Furthermore, there is some concerns that the multiplicity of questionnaires, the time necessary for completing them and especially the fact that if it is too long the participant might answer inappropriately. How does the author address this specific issue?

 R/ Thanks for this comment. Once an eligible patient is identified and that his/her consent is 

 obtained for the study, physical/anthropometric measurements will be taken. After that, the 

 questionnaire is administered by a data collector. At the end, the blood sample is taken. 

 See Lines 272-275

 It is estimate that the time to complete the questionnaire would be approximately up to 60 

 minutes. 

 See Line 274

 The participants will be informed of the details of the study, issues of reimbursements 

 and will only participate once done with other clinic activities or while waiting to be attended.

 Line 196-198

• Line 237, would recommend adding the current working status

 R/ Thanks for comment. The current working status is one of the study’ variables of the study as 

 it can be verified in the data collection tool. It was unfortunately forgotten in the list of 

 variables. The list of the variables has been reviewed to include all the information sought from 

 participants.

 See S2 Appendix, page 2, question 7

 The list of the variables has been reviewed to include all the information sought from 

 participants.

 See Line 280 and named occupation

• No information is given regarding who will drive the Delphy consensus data collection and interpretation.

 R/ Thanks for this comment. A research team lead by the Principal Investigator and including at least one 

 supervisor and a statistician will conduct data collection and interpretation of the data. The information has been 

 added to the protocol.

 See Lines 449-451

• No information is given regarding the research assistant’s integration in the authorship.

 R/ Thanks for the comment. The research assistant will be acknowledged for their support in the study. For the 

 authorship, we will follow guidelines from the International Committee of Medical Journal Editors (ICMJE). I added 

 the above information in the protocol.

 See Lines 525-528, Reference 51

• Is the retribution of the participants specifically specified before inclusion? It might obviously induce some bias

 R/ The retribution for the participants is described in the informed consent form. A reimbursement of the transport is 

 only given when the patient is invited for interviews or administration of the questionnaire.

 See Informed consent form for patients in sub-study 1, page 5, Payment

• How long is an interview planned to last? Is there a timeframe for the interview?

 R/ There is no timeframe for the interviews. For the qualitative phase, it is supposed to last up to 45 minutes. And for 

 the quantitative phase, the administration of the questionnaire will take approximately no more than 60 minutes. This 

 information is added in the protocol. This information can also be seen in the patient information sheet.

 See Lines 206-207, 274; Informed consent form for patients in sub-study 1, page 2

Reviewer#3

• Although the manuscript recognizes the importance of restructuring health systems with greater empowerment of patients, involvement of their families and the communities in the care, the perspectives of family members are not taken into account in this study.

 R/ Thanks for this comment. Family members have not been considered for interviews. We 

 recognize that could be of great help to have a broader view. Nevertheless, in this particular 

 study, we have chosen the healthcare providers and the patients as it was showed that their 

 perspectives could help build an adequate diabetes care system (Pun et al.,2009).

 See Lines 114-115

 Reference: Pun SP, Coates V, Benzie IF. Barriers to the self-care of type 2 diabetes from both 

 patients’ and providers’ perspectives: literature review. J Nurs Healthc Chronic Illn . 2009 Mar 

 ;1(1):4–19.

• There is a high risk for selection biases, which have also been described in the manuscript.

 R/ Thanks for the comment. Selection bias mainly concerns the qualitative component of the study. The purpose of 

 the study is to describe the participants’ perspectives on ways to improve glycaemic control. We assume that we will 

 have advantage to find out information-rich patients about the glycaemic control. And that the purposive sampling 

 approach described will ensure to reach them. 

Abstract

• I would suggest presenting the numbers on T2DM patients with poor glycaemic control in

Kinshasa instead of reporting ‘a large proportion of patients’.

 R/ Thanks for the comment. We have added estimated prevalences of poor glycaemic control in our setting (as 

 described in previous studies).

 See Line 40

• I would suggest providing more details on the qualitative study as part of sub-study 1 (e.g. in-depth review).

 R/ We are grateful for the comment. We have added more details of the sub-study 1 in the 

 abstract. The revised description stated that “A minimum of 20 purposively selected patients will 

 participate in the qualitative study that will involve in-depth interviews about their perspectives 

 on glycaemic control”. Therefore, the sampling method, the data collection tool and the objective 

 are added. 

 See Lines 49-50

 The data analysis is also described.

 See Lines 52-53

Introduction

• For non-expert readers in the field: could you elaborate on/provide more details or examples of the adjustments patients with diabetes are required to make, as well as on the definition of glycaemic control (this is not mentioned until the methodology part of the paper (HbA1c < 7%))?

 R/ Details on the adjustments have been included in the text. 

 See Lines 71-73 and Reference 5

 Definition of glycaemic control included in the introduction. 

 See Lines Lines 85-87

• 50% of patients achieve glycaemic control worldwide. Does this number also include T1DM?

 R/ This number only consists of patients with type 2 diabetes as specified in the text.

• Could you provide more details on why poor glycaemic control in T2DM patients varies across settings and are complex?

 R/ Details have been added between parentheses for each of the characteristics

 See Lines 77-82

• It is unclear whether the reported prevalence of diabetes in the DRC (4.8%) includes both T1DM and T2DM.

 R/ We are grateful for this comment. The prevalence of 4.8% reported if for both forms. To be 

 more specific, we have added that the percentage of T2DM in the diabetes population is about 

 92% as found in one study. The reference has been added. 

 See Line 85

• In the following sentence, it is not clear to me what 'the development' refers to: ‘The burden of diabetes is a real hindrance to the development of both patients and their families, and also of the health system oriented towards other priorities with limited resources’

 R/ Thanks for this comment. For better understanding, the sentence has been changed as follows: ‘The burden of 

 diabetes is a real hindrance to the wellness of both patients and their families, and the effectiveness of the health 

 system oriented towards other priorities with limited resources’

 See Lines 91-93 

Material and methods

Questions related to the qualitative phase

• The manuscript describes the interview of 20 patients until thematic saturation is reached. Patients are selected purposively with both good and poor glycaemic control. Will a blood sampling be performed before the interview to determine the HbA1c level, or how is the glycaemic control prior to the interview defined? How is this number (20 interviews) distributed across the patients with good and poor glycaemic control?

 R/ We are grateful for the comments and questions raised. For the qualitative component of the sub-study 2, patients 

 will be selected independently of their glycaemia control status. A blood sample will be taken from participants for 

 HbA1c assay. But this exam will not be available the same day. We will report on the distribution of poor versus good 

 glycaemic controlled patients in the sample. 

 See Lines 207-209

• Is it necessary to interview T2DM patients with good glycaemic control, since the research objective is to define drivers for poor glycaemic control? Or is another objective to define drivers for good glycaemic control as well?

 R/ Yes, it is necessary to interview both good and poor controlled patients on drivers of poor 

 glycaemic control. Getting the perspectives from these two groups will help to broader the view 

 on the phenomenon. We assume that the control of glycaemia is not definitive and even controlled 

 patients have been uncontrolled at one point of their disease and could contribute to the 

 understanding of poor glycaemic control. 

• Is there a difference made between patient with treatment < / > 7 years (cf. quantitative phase)?

 R/ Yes, a difference will be made between patients according to treatment duration. 

 We have added a variable for duration treatment in the questionnaire.

 See S2 Appendix, page 7, section 23e

• Are the interviews performed in 1 of the 66 centers (which one(s))?

 R/ No, the interviews will not be conducted in 1 center. These will be conducted in the centers selected for this study. 

 Their total number is 20.

 See Lines 174-177

• How much time will the interview approximately take?

 R/ The interview will approximately take up to 45 minutes. The information has been added in 

 the manuscript and in the Informed consent form for patients in sub-study 1

 See Lines 206-207 and Informed consent form for patients in sub-study 1, page 2. 

Questions related to the quantitative phase

• The protocol describes an interview procedure, which, in my opinion, is confusing since 

 validated questionnaires are used.

 R/ Aligning to reviewer’ comment, we will use the term “administration of the questionnaires” 

 instead of “interviews”.

 See Lines 258, 271-274

• Since only 20 centers are selected to recruit patients, it seems to me the multicentric study 

 does not comprise 66 centers.

 R/ No, the multicentric study will not comprise all the 66 centres. We will use sampling 

 proportional by the size of diabetic patients attending the healthcare facilities to select 20 centres 

 where the study will take place.

• Is there a specific reason why the participants will not fill in the questionnaires themselves, 

 but the researchers ask the questions? Will this not create an interpretation bias? Can the 

 questionnaires be completed online by the participants in their home situation?

 R/ Considering our setting, very few patients have access to email or the internet to be able to 

 complete the questionnaire through online. This will also introduce a selection bias, if used. The 

 level of education of our patients is also a big hindrance to self-administration of the 

 questionnaire by patients. There is a risk of interpretation bias, which has been identified and will 

 be prevented by the training of data collectors.

 See Line 387

• Why are the characteristics of diabetic disease not collected from the patient’s medical file?

 R/ The characteristics of diabetic disease are primarily sought from the participants. We use 

 patients ‘medical to verify the information given. 

 See Lines 288-289

• How much time will the completion of the questionnaires take? It seems a lot of information is 

 asked (18 pages); is this all necessary information related to the research objective?

 R/ The time to complete the questionnaire will approximately be of no more than 60 minutes. 

 Once the data collector is familiar with the questionnaire, the time to complete it is shortened. All 

 the information sought is related to the research objective.

 See Line 274

 All the information sought is related to the research objective.

 Sub-study 2

• Patients will be selected if they have been followed for at least six months for their diabetes 

 and if they have integrated the model of care offered to patients with diabetes. It is not clear 

 what this ‘model of care’ comprises.

 R/ A model of care broadly defines the way health services are delivered. It comprises the 

 caregivers (doctors, nurses), process of care (diagnostic, treatment, follow-up), access to care, and 

 availability of resources.

 OTHER

• We will use SATA 17 rather that STATA 16 for data analysis (Line 371)

• List of references updated with inclusion of new references (References number 5, 26, 30, and 51)

• One more element added for enhancing the credibility of the study (Lines 223-224)

• More details are added for the selection of patients in the qualitative phase of the sub-study 1 (Lines 183-186)

• More details have been provided about data analysis in qualitative component of the sub-study 1 (Lines 215-218)

• More details are provided in the selection of the patients and healthcare providers in the sub-study 2 (Lines 412-415)

• More details have been provided about data analysis in the sub-study 2 (Lines 432-436)

• Changes have been made in the Authors contributions section (Lines 566-573)

• The quantity of veinous blood to collect is 2 millilitres instead of 5 millilitres. This information appears in the manuscript and the Informed consent form for patients in sub-study 1 (Line 359; Informed consent form for patients in sub-study 1, page 2).

• Language editing for grammatical errors.

---

## [Decision Letter · Decision Letter 1]

25 Apr 2022

Protocol: Developing a framework to improve glycaemic control among patients with type 2 diabetes mellitus in Kinshasa, Democratic Republic of the Congo

PONE-D-21-29300R1

Dear Dr. Fina Lubaki,

We’re pleased to inform you that your manuscript has been judged scientifically suitable for publication and will be formally accepted for publication once it meets all outstanding technical requirements.

Kind regards,

David Desseauve, MD, MPH, PhD

Academic Editor

PLOS ONE

Additional Editor Comments (optional):

Reviewers' comments:

Reviewer's Responses to Questions

**Comments to the Author**

1. Does the manuscript provide a valid rationale for the proposed study, with clearly identified and justified research questions?

Reviewer #1: Yes

Reviewer #2: Yes

Reviewer #3: Yes

2. Is the protocol technically sound and planned in a manner that will lead to a meaningful outcome and allow testing the stated hypotheses?

Reviewer #1: Yes

Reviewer #2: Yes

Reviewer #3: Yes

3. Is the methodology feasible and described in sufficient detail to allow the work to be replicable?

Reviewer #1: Yes

Reviewer #2: Yes

Reviewer #3: Yes

4. Have the authors described where all data underlying the findings will be made available when the study is complete?

Reviewer #1: Yes

Reviewer #2: Yes

Reviewer #3: Yes

5. Is the manuscript presented in an intelligible fashion and written in standard English?

Reviewer #1: Yes

Reviewer #2: Yes

Reviewer #3: Yes

6. Review Comments to the Author

You may also provide optional suggestions and comments to authors that they might find helpful in planning their study.

Reviewer #1: All my previous comments have been adressed. I have got no more comment and I thanks the authors for this submission.

Reviewer #2: I would like to thanks and congratulate the authors for their extensive revision of the manuscript, which has significantly improved.

Reviewer #3: The authors provided clear and structured answers to the review comments and made adjustments where applicable. There are no additional comments to the authors.

7. PLOS authors have the option to publish the peer review history of their article (what does this mean?). If published, this will include your full peer review and any attached files.

Reviewer #1: No

Reviewer #2: No

Reviewer #3: No

---

## [Editor Report · Acceptance letter]

9 Jun 2022

PONE-D-21-29300R1 

Protocol: Developing a framework to improve glycaemic control among patients with type 2 diabetes mellitus in Kinshasa, Democratic Republic of the Congo 

Dear Dr. Fina Lubaki:

I'm pleased to inform you that your manuscript has been deemed suitable for publication in PLOS ONE. Congratulations! Your manuscript is now with our production department. 

Kind regards, 

on behalf of

Dr. David Desseauve 

Academic Editor

PLOS ONE